# Pharmacy Practice in High-Volume Community Settings: Barriers and Ethical Responsibilities

**DOI:** 10.3390/pharmacy9020074

**Published:** 2021-04-03

**Authors:** Christopher T. Owens, Ralph Baergen

**Affiliations:** 1Kasiska Division of Health Sciences, Idaho State University, Pocatello, ID 83209, USA; 2Department of Philosophy, Idaho State University, Pocatello, ID 83209, USA; baerralp@isu.edu

**Keywords:** community pharmacy, ethics, job satisfaction, pharmaceutical care, public health, retail, Code of Ethics, pharmacy practice

## Abstract

Pharmaceutical care describes a philosophy and practice paradigm that calls upon pharmacists to work with other healthcare professionals and patients to achieve optimal health outcomes. Among the most accessible health professionals, pharmacists have responsibilities to individual patients and to public health, and this has been especially evident during the COVID-19 pandemic. Pharmacists in high-volume community settings provide a growing number of clinical services (i.e., immunizations and point-of-care testing), but according to job satisfaction and workplace survey data, demands related to filling prescriptions, insufficient staffing, and working conditions are often not optimal for these enhanced responsibilities and lead to job dissatisfaction. Professional codes of ethics require a high level of practice that is currently difficult to maintain due to a number of related barriers. In this paper, we summarize recent changes to the scope of practice of pharmacists, cite ethical responsibilities from the American Pharmacists Association Code of Ethics, review data and comments from workplace surveys, and make a call for change. Corporate managers, state boards of pharmacy, and professional organizations have a shared responsibility to work with community pharmacists in all settings to find solutions that ensure optimal and ethical patient care. Attention to these areas will enhance patient care and increase job satisfaction.

## 1. Introduction

Patient-centered pharmaceutical care describes a philosophy of enhanced responsibility for pharmacists to work toward optimal health-related outcomes of patients whom they serve [1]. According to the Principles of Practice for Pharmaceutical Care put forth by the American Pharmacists Association (APhA), this ideal describes “a patient-centered, outcomes-oriented pharmacy practice that requires the pharmacist to work in concert with the patient and the patient’s other healthcare providers to promote health, to prevent disease, and to assess, monitor, initiate, and modify medication use to assure that drug therapy regimens are safe and effective” [2]. More than simply dispensing medications from a valid physician’s order—as has been the historical role of the pharmacy profession [3]—pharmacists who practice under this paradigm consider themselves active members of the healthcare team and partners with physicians, nurses, other healthcare professionals, as well as with their patients, and they place particular emphasis on their own area of expertise: the appropriate use of medications. They are likewise proactive in ensuring that patients are prescribed and are adherent in taking the best medications for their given conditions at the right dose, via the right route, for the right duration, and with a clear understanding of the known benefits and risks.

Pharmacists are also being increasingly recognized for their role as front-line healthcare professionals with more direct and regular access to patients than other members of the healthcare team, with the majority of Americans living within five miles of a community pharmacy [4]. Pharmacists continue to be an important resource for patients in need of health-related information and are well positioned to provide preventative health screenings and services that historically have been in the purview of physicians and other primary care providers [5].

When this higher-level ideal of patient-centered pharmaceutical care was first introduced three decades ago, it was met with a mixture of excitement and trepidation on the part of the pharmacy community. On the one hand, it clearly defined a mission and future for the profession of pharmacy and was to act as the philosophical basis for enhanced professional responsibility as well as elevation in professional status. It is likely that for this reason, it was quickly enshrined as the central feature of the Code of Ethics for Pharmacists by the APhA, which was adopted in 1994 (Figure 1: APhA Code of Ethics) [6]. On the other hand, this new paradigm called for a radical shift from the status quo of traditional pharmacy practice and represented a role some believed the profession would be unable to adopt fully due to a variety of practice-setting barriers, including necessary changes to pharmacy staffing and layout, employer attitudes, and patient perceptions [7,8]. The COVID-19 pandemic has brought into clearer focus the need to better identify and overcome these barriers in order for pharmacists to live up to their ethical and professional mandate and to fully fill their vital roles [9]. In this paper, we will briefly explore the changing landscape of community pharmacy, the expanding scope of practice of pharmacists, the current working conditions in community pharmacies, and the barriers they present to patient care; then, based on the current Code of Ethics, we outline a mandate for pharmacists in busy community settings to engage with corporate managers, professional organizations, boards of pharmacy, and other relevant stakeholders to find solutions to the issues that face day-to-day community practice and help community pharmacists better fulfill their professional roles, enhance their own job satisfaction, and improve patient care.

## 2. Community Pharmacy Settings

While the practice of pharmacy is diverse and performed in many different settings ranging from hospitals and clinics to rehabilitation and long-term care centers, the largest group and most visible members of the profession are pharmacists working for commercial chain drug, department, or grocery stores—a group that represents approximately 60% of pharmacists in the United States [10]. The perception of pharmacists by the lay public, as well as by physicians, nurses, and other healthcare professionals, is largely shaped by their interactions with community pharmacists, and indeed, this segment of the pharmacy profession has long found itself vying for approval from its peers in hospitals and other clinical settings, and it has had to fend off outdated allegations of “incomplete professionalization”—the notion that pharmacists are unlike other medical professionals because of significant constraints on their autonomy related to a perceived lack of professional responsibility [11]. While more and more community pharmacists routinely provide drug consultations, take patient health histories, give immunizations, perform basic physical assessments, and provide other clinical services, pharmacists in these settings are still not often recognized for these roles by other healthcare professions nor do they at times even give themselves sufficient credit for performing them [12].

The practice of pharmacy in community settings has changed dramatically in the past two decades in the U.S. as well as internationally, with pharmacists now providing services that go well beyond the filling of prescriptions and providing patient counseling on prescription and over-the-counter medications [5]. Most often working in high-volume retail settings, community pharmacists are increasingly having to find ways to balance their obligations to their corporate managers as well as to their patients while working under highly stressful conditions and often without adequate staffing [13,14]. Such obligations can stretch pharmacists to their limits as they must keep up with corporate demands and a variety of productivity metrics, and at the same time perform their professional roles: giving due diligence to confirming the appropriateness of prescriptions, identifying and solving potential drug-related problems, educating patients about their medications, and providing a growing number of clinical services.

## 3. Clinical Services in Community Pharmacies

Although the fast-paced atmosphere of retail settings continues to gain speed and prescription volumes continue to increase, community pharmacists provide a growing number of clinical services, including Clinical Laboratory Improvement Amendments (CLIA)-waived point-of-care (POC) tests, such as for blood glucose, hemoglobin A1c, and cholesterol [15]. State laws and scopes of practice are continually changing in response to demands for these and other similar services. Giving immunizations, including the seasonal flu vaccine, is now commonplace in virtually all community pharmacies, and in many U.S. states, pharmacists, student pharmacists, and pharmacy technicians may provide this and other vaccinations as well, including for pneumococcus and herpes zoster. Since the H1N1 pandemic of 2009, a marked shift has been noted with respect to where vaccinations are taking place, with a move from physicians’ offices to pharmacies [16]. Other clinical services routinely performed in community settings include the provision of medication therapy management (MTM), an extensive individualized review of drug therapy in an effort to optimize treatment and reduce costs [17]. In an increasing number of U.S. states, pharmacists may start, stop, or modify drug therapy under collaborative practice agreements and in some cases, independent prescribing of certain medications is permitted, including statins for patients with diabetes, inhalers for patients with asthma, and antibiotic treatment for patients presenting with uncomplicated urinary tract infections [18,19]. All of these changes have increased the clinical role of pharmacists, including those working in high-volume community pharmacy settings.

## 4. COVID-19 Pandemic

The COVID-19 pandemic has put enormous pressure on already strained healthcare resources and personnel and has also highlighted an increased role for community pharmacists—not only as part of the national COVID response, but for other services that are being forgone by some patients due to decreased access to or avoidance of primary care and other healthcare settings [9]. In July of 2020, by executive order, the U.S. Department of Health and Human Services (HHS) authorized pharmacists to give childhood vaccinations as part of the Public Readiness and Emergency Preparedness (PREP) Act. In addition, pharmacists may also order and administer COVID diagnostic tests [20,21]. In other countries, similar changes in practice and day-to-day expectations were also observed. A survey of community pharmacies in Canada showed surges in volumes of patients seeking prescription renewals, increases in drug shortages, increased workload with respect to dispensing-related policies designed to prevent stockpiling, and increases in unremunerated clinical interventions. Lack of pharmacy-specific guidance in dealing with COVID-related issues often resulted in the need for planning to be devised at the local level, which included longer shifts, different scheduling and staffing patterns, and enhanced reliance on technology. All of these contributed to new daily stresses [22]. Although their workload has greatly increased, community pharmacists have largely answered the call to provide these additional services, and while there are clear indications of willingness to fill these needs, significant barriers continue to be identified—barriers that preceded COVID-19 and that will likely continue after it.

## 5. Potential Barriers

Even before the COVID-19 pandemic, pharmacists working in high-volume retail settings commonly faced significant barriers to fulfilling their expected professional obligations, including high stress, inadequate staffing, and insufficient time. Workplace surveys are often used to better quantify these factors, including their perceived severity and impact, and while definitive conclusions must be drawn with caution from surveys with lower response rates, important information may still be gleaned from such data sources. Findings from these surveys have consistently highlighted issues related to job satisfaction and potential contributors to burnout—as well as potential ethical considerations [23,24,25,26].

A salary and job satisfaction survey is conducted annually by *Pharmacy Times*, a clinically-based monthly journal in the United States that provides practice-related information for community, specialty, and health-systems pharmacists. In these surveys, pharmacists are asked to report their overall job satisfaction on a scale of 1 to 7 (with 1 being “not at all” and 7 being “extremely”). In 2020, the average response of the 299 pharmacists was 4.53. The top reasons provided for dissatisfaction were the same as those from previous years and included workload, management-related issues, and work/life balance. It was also reported that respondents who said they were dissatisfied did not want to comment on the record [23].

An annual survey conducted in 2019 by the Midwest Pharmacy Workforce Research Consortium found that over 70% of respondents rated the workload at their primary place of employment as “high” or “excessively high”, which was slightly higher than ratings in previous years (66% and 68% in 2014 and 2009, respectively). The highest proportions of pharmacists who rated their workload as “excessively high” or “high” were in chain (91%) and mass merchandiser (88%) settings [24]. Increased stress was reportedly related to inadequate staff support, increased paperwork, and perceptions of a negative work environment.

A working conditions survey completed for the Oregon Board of Pharmacy in 2011 found similar results [25]. This survey was conducted in response to reports of understaffing and inappropriate working conditions and was sent to 4800 licensed pharmacists in the state. A total of 1400 responses were received for a response rate of 29%, half of whom were pharmacists in chain/mass merchandiser settings and another 30% reported working in independent or “other retail settings”. Less than half of respondents indicated that their “working environment was conducive to providing safe and effective patient care” and less than half agreed that their pharmacy had “adequate pharmacist staff to provide safe and effective patient care”. It was noted that the majority of the 514 comments received (77%) were critical of their working conditions, and most of these comments were from pharmacists in non-independent, large chain retail settings. One of the common themes noted by the survey reviewers was that “respondents presented a picture of an environment in which it appears the interests of the corporations making a profit are not aligned with the interests of the pharmacists and, at times, the patient.” It was also noted that along with decreasing staffing levels, there has been an increase in corporate expectations of pharmacists related to the provision of patient care services, “such as immunizations, counseling, medication management, and blood pressure monitoring in addition to all of the requirements for properly filling prescriptions.” A smaller group of respondents highlighted “ergonomic issues” related to the physical layout and work environment in their pharmacies, “expressing concern about noise levels making it difficult to concentrate [and] inadequate privacy for counseling patients”.

A survey conducted among Serbian community pharmacists identified many of the same issues reported in America [26]. Over 700 pharmacists from 23 pharmacy chains were asked their perceptions related to the impact of typical daily workplace conditions on patient care. Pharmacists were asked to report the frequency of specific problems arising in their practice using a Likert scale (1 = does not occur, 2 = rarely occurs, 3 = sometimes occurs, 4 = usually occurs, and 5 = always occurs). With a response rate of almost 80%, pharmacists reported the following: A pharmacist performs several tasks simultaneously, while providing pharmaceutical services to a patient (median 3.51 +/− 1.08); A pharmacist needs to provide confidential information to the patient when the patient’s privacy is compromised by the presence of other patients (3.56 +/− 1.08); A pharmacist is under pressure to achieve daily sales targets for the pharmacy (3.54 +/− 1.22). Approximately 7% of pharmacists in the survey reported that they have had to compromise their ethical values to meet a supervisor’s or employer’s request.

In a recent survey conducted in the State of Idaho in the United States, licensed pharmacists were asked about their willingness to provide COVID-19 related services, as well as their perception of potential barriers, and it found similar issues in terms of a disconnect between what is being required of them and the time and staffing they have to be able to fulfill these expectations [27]. The survey was completed by 229 pharmacists for a 13.6% response rate, with almost half of the responses received from pharmacists in community settings (chain drugstore, supermarket, mass merchandiser, and independent pharmacies). The findings indicated that approximately three-quarters of respondents were willing to provide COVID testing and similar services and more than 80% would be willing to prescribe and administer vaccines when available; however, several comments also included significant barriers in terms of workflow, logistics, and lack of appropriate reimbursement models for these additional services. One comment summed up the sentiments of many in this way: “We already work with not enough man hours to do our jobs. This would be just another metric to be measured by corporations and we would have no choice in the matter and be required to do it. We are already stretched thin, especially during the flu season.”

## 6. Business vs. Ethics

The day-to-day realities illustrated by these survey findings point out a major ethical failing inherent in the high-volume community pharmacy setting: community pharmacists often lack sufficient time or appropriate workspaces to fulfill their Code of Ethics required responsibility to “help individuals make the best use of medications.” Any pharmacist scrupulously performing all of the safety checks, educating patients on appropriate use of drug therapy, explaining potential side effects, double-checking with prescribers when the need arises, and providing the range of clinical services now offered, may have extreme difficulty keeping up with the pace. This is especially true in understaffed pharmacies. The related barrier of inadequate physical layout of many community pharmacies is also a concern, as an appropriate place for truly private and confidential discussions with patients is sometimes lacking. While this is changing in many community settings, pharmacists do find themselves discussing illnesses, medication side effects, and entitlement programs, along with giving immunizations and providing POC testing and other clinical services, in places where it can be difficult to fully ensure patient comfort and privacy and encourage meaningful healthcare professional/patient dialog.

The disconnect between the ideal of patient-centered pharmaceutical care, with all the benefits that can result from it, and the typical day in a high-volume community pharmacy setting can be frustrating and disappointing—as illustrated by workplace survey findings [23,24,25,26]. Even more importantly, the situation is also unethical, and it needs to be changed. In working to improve matters, it is valuable to understand the conflicts that arise when standard business models and practices are applied in healthcare settings. While most pharmacists are acutely aware that they are healthcare professionals first and retailers second, the conditions of their employment sometimes exact a necessary change in priorities to accommodate the demands of their employers’ corporate focus.

The first—and, by implication, the highest—principle in the APhA’s Code of Ethics for Pharmacists is this: “A pharmacist respects the covenantal relationship between the patient and pharmacist” [6]. In explicating this, the Code states, “a pharmacist promises to help individuals achieve optimum benefit from their medications, to be committed to their welfare, and to maintain their trust.” Pharmacy, similar to the other branches of healthcare, strives to be patient-centered. The Code is explicit about this: “A pharmacist places concern for the well-being of the patient at the center of professional practice.” This is the yardstick by which policies and practices must be measured. When the drive to fill unreasonably high numbers of prescriptions or to maintain particular revenue levels prevent pharmacists from acting in accordance with their profession’s guiding principle, then those practices are flawed and must be reconsidered.

The second and third principles in the APhA’s Code of Ethics emphasize the importance of maintaining confidentiality and respecting patients’ dignity [6]. Here, too, the reality of many high-volume community settings falls short of the ideal. These principles are a central, driving force behind the concept of patient-centered pharmaceutical care [1]. In order to promote patients’ well-being, pharmacists must be able to adequately review drugs and dosages, check for interactions, educate patients on their drugs’ proper use and inform them of potential side effects, and have the ability to confer with the prescriber should the need arise. When this is done properly, the benefits to patients can be substantial. Adverse drug events are prevented [28,29], medication errors are identified and corrected [30,31,32], and patient outcomes are improved [33,34,35]. Furthermore, in the United States, federal law requires that pharmacists counsel patients appropriately on all dispensed medications [36]. Clinical services provided by pharmacists in community settings must likewise be able fit into the standard of practice workflow, as they represent another part of the pharmacist’s ethical responsibility to serve their patients’ needs and have been shown to result in substantial public health benefits in different countries around the world [37,38,39,40].

Law, professional ethics, patients’ well-being, and public health all indicate that pharmaceutical care ought to be carried out properly, and yet pharmacists in many community settings are unable to do so. Why has the retail setting strayed so far from what it ought to be? It is tempting to say that the desire to maximize profits is to blame, but this oversimplifies the problem. It would be more accurate to say that the business model that has been applied to community pharmacy was not designed to accommodate the nature of the pharmacist/patient relationship nor to facilitate the type and level of clinical interaction that is now the expectation for many of these encounters.

When the pharmacist is understood as a retailer, the patient becomes a customer, and their interactions are understood in mainly economic terms. This is likely the root of the problem. In general, retailers have no particular obligation to promote the well-being of their customers. Retailers are not expected to train customers in the proper use of products, to ensure that the most appropriate products are being purchased, or to warn of any ill effects that might result from their use. Furthermore, retailers do not offer preventative health screenings and related clinical services that promote individual and public good. These are instead professional roles. However, it is important to note that pharmacies’ corporate management may view these interactions predominantly from an economic perspective. Their goal is to employ the pharmacist who has the expectation of maximizing revenue and minimizing the cost of each customer interaction. The guiding considerations are not necessarily focused on patient well-being, but rather upon competitiveness with other pharmacies and return on investment.

Given the current retail environment in which many community pharmacies are situated, it is understandable that the pharmacy counter may be viewed as not so very different from other retail outlets. While the store’s management must accommodate the various legal requirements for proper product storage and record-keeping associated with a pharmacy, this is certainly not unique to the pharmacy department: special requirements must also be met at jewelry and deli counters. It is also understandable that when a store’s management decides to add a pharmacy, they may not fully appreciate all of the professional implications that come with entering into the realm of healthcare, which is something that differs profoundly from the other departments in the store. If the same business models are applied, with the main focus on revenue, investment, and efficiency, then something truly fundamental may be inadvertently neglected.

## 7. Conclusions

The conflicts between business and healthcare models are unavoidable. The relationship between patients and medical professionals will always involve very different components and obligations than are found in the customer/retailer relationship. Fulfilling the medical professional’s obligations to patients will often cost money at the time of implementation and may appear to or actually detract from maximal efficiency; therefore, managers who are trying to carry out the strategies they were taught in business school may tend to resist efforts to spend more time, money, or energy on activities necessary for appropriate patient care (i.e., different physical layouts and increased staffing). At the same time, pharmacy staff job satisfaction and the human costs associated with failure to remedy the current situation likewise has economic impact.

Although the problems outlined here are understood by pharmacists in community settings and have grown in recent decades, little attention has been paid to implementing sustainable ways to resolve them. While dialog such as this is a beginning, it is only the start of a much bigger project in medical, professional, and business ethics. There is a major role for corporate managers to play in addressing pharmacy layout, staffing, and working conditions in order to better support community pharmacists as they fulfill their professional roles, especially in high-volume retail settings. Managers may be greatly aided in dealing appropriately with these issues by being more inclusive of front-line practicing pharmacists in planning and implementation of new services or when setting workplace expectations. For example, an analysis from the English National Health Service (NHS) showed better clinical outcomes when clinicians were included on boards of directors of hospital trusts [41]. Likewise, there is a role to be played by state boards of pharmacy, as their primary responsibility is “to support patient and prescription drug safety” and to safeguard public health [42]. There is still a further role to be played by state, regional, and national pharmacy professional organizations, such as the APhA, and analogous organizations in other countries. Such organizations exist to “advance the entire pharmacy profession” and to advocate for changes in practice that benefit individual patients, the healthcare system in general, and practicing professionals [43].

Pharmacy’s professional identity and daily practice is so closely linked with the constraints of the business world that this interconnectedness must be further explored. A combined approach that involves all the key stakeholders, including individual pharmacists, corporate managers, state boards of pharmacy, and professional organizations will be needed to better characterize and address each of the identified barriers appropriately—this is a critical ethical mandate for the profession, and these problems must be resolved before the ideal of pharmaceutical care and all of its benefits to patients and public health can be fully realized.

## Figures and Tables

**Figure 1 pharmacy-09-00074-f001:**
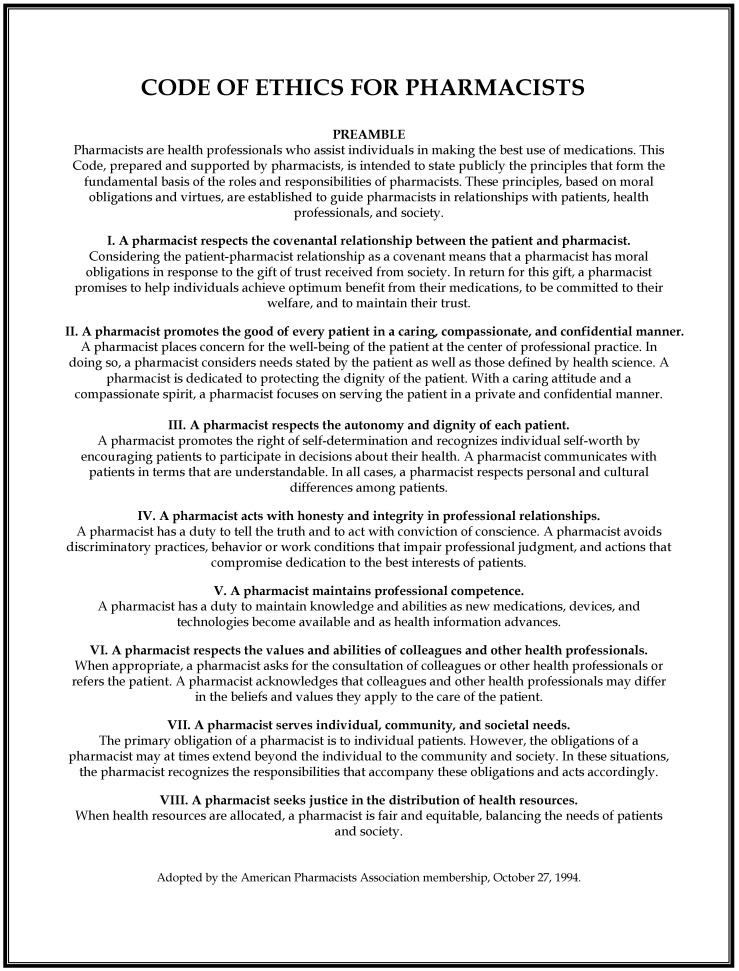
American Pharmacists Association (APhA) Code of Ethics (https://www.pharmacist.com/code-ethics?is_sso_called=1, accessed on 20 November 2020).

## Data Availability

Not applicable.

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
