# Peer review of "Pharmacy Practice in High-Volume Community Settings: Barriers and Ethical Responsibilities"

_pharmacy, 2021, doi:10.3390/pharmacy9020074_

Round 1

Reviewer 1 Report

Thank you for allowing me to review this manuscript.

Abstract:
Your purpose needs to be reworded: you summarized recent changes, cited code of ethics, reviewed few data resources, but you did failed to provide any recommendations.

2. Community Pharmacy settings:

Line 77. Need to explain what you mean here by "incomplete professionalization"

3. Clinical Services in Community Pharmacies

Line #113 says "start" twice

Line #118 Consider rewording, this seems redundant. "High-volume community settings"

4. COVID-19 Pandemic

Need to include how this affected workflow (positive/negative/no impact)

5. Potential barriers

Is this barrier to the COVID 19 changes or overall? I would consider separating those out

Line #146 The conclusion from these results is being treated as significant although the survey has a 29% response rate which is normally not considered significant. Most standards for a significant response rate is 60%. 

Line #168 Again, this is a 13.6% response rate. There needs to be some addressing the limitation of the results. Otherwise it reads as selection bias.

6. Business vs. Ethics

Line 183 spell out "isn't" to "is not"

Line 185 avoid cliches like "buy the book"

Line 196 "idea" not "ideal"

Line 203 this is biased. Do you have a reference that the priorities are "begrudged"?

Paragraph 5, 6, and 7 under this section have biased language.

Avoid words like "merely, extremely, no wonder, hypothetical questions"

Line 253 and 258 you use "no wonder" twice in same paragraph. This shows bias

7 Conclusion

Line 274 use an objective word instead of "good"

Line 280 "valuable" is biased. let the reader decide if it is valuable. Unless you have a reference showing that this conversation is needed.

Overall

This manuscript seems like "a whole lot of talk" with no solutions proposed. I would like to see more about error rates or average wait times or something to show concrete "pharmacists can’t do their jobs properly” type numbers, but that might not be available. All data presented were about perceptions.

Current working conditions mean low job satisfaction, inability to follow code of ethics due to inability to practice safely and deliver patient care. However, this could have expanded on the “pharmacy being primary care” point - i.e. phone call volume up due to people are calling for medical advice afraid to go to ER or doctors office, telehealth has made it impossible to speak to doctors, so poor clinical outcomes because (a) patients can’t reach doctor and (b) delayed prescriptions because pharmacists can’t reach doctor for clarification. Adding COVID-19 vaccines has turned hours-long wait times into days-long wait times.

This study has strong potential but needs more objective evidence to help reduce bias.

Author Response

Thank you for your comments. Please see the attached document with our responses using the track changes function in Word. 

Reviewer 2 Report

Thank you for the opportunity to review a commentary on about an ongoing conflict between pharmaceutical care and retail pressures.  The article is very US focused and perhaps would benefit from a more international perspective for an international journal.  There are many unsubstantiated claims that would be the authors’ opinion.

The claims made in the introduction would benefit from being supported by existing literature, rather than authors opinion.  References are required at the end of the first and second paragraphs and at the end of the sentence that finishes in line 54. 

Line 56, Is pharmaceutical care a ‘new’ paradigm when it has been discussed since 1994?

Line 59-60: mentions practice-setting barriers, what barriers are these?

Line 66  the claim that pharmacists are “essential front-line health care professionals” is not supported by the introduction.

Lines 71-73 appear to be about community pharmacists but don’t say so.

Ref 9 is from 1968, is there anything more recent?

Line 113  double up of start,

Line 131  “significant barriers have also been identified”  How were they identified, and by whom?  What are these barriers?

The Potential barriers section is study rather than content focused with a paragraph each for 3 different studies.

The first two paragraphs in section 6 Business vs ethics are not supported by any reference to the literature.  Neither are paragraphs 5, 6 or 7 of this section.

Authors opinion supports the conclusion but little evidence from the literature has been included that provides support for conclusions.

The reference list titles are a mixture of title and sentence case.

Author Response

(The authors gave the same response as above.)

Reviewer 3 Report

Manuscript ID: pharmacy-1151388

Title: Pharmacy Practice in High-Volume Community Settings: Barriers and Ethical Responsibilities

Pharmacists have a great responsibility to individual patients and to public health which has been especially evident during the COVID-19. Therefore, authors reviewed recent changes to the scope of practice of pharmacists, cited ethical responsibilities from the American Pharmacists Association Code of Ethics, reviewed data and comments from workplace surveys, and provide recommendations and a call for change.

Overall, the manuscript is well written. Some improvement will increase the readability of the paper.

  1. The findings on barriers and ethical responsibilities can be presented in table format.
  2. Figure 1 can be shared using supplementary file since it is commonly available online, and authors have no contribution to this content.
  3. One of the objectives “provide recommendations” which need to address clearly and authors can provide them parallelly with the corresponding barriers in the same table.
  4. The reference is well-formatted.

Author Response

(The authors gave the same response as above.)

Round 2

Reviewer 1 Report

Thank you for your changes.